# Spectrum and Prevalence of Rare *APOE* Variants and Their Association with Familial Dysbetalipoproteinemia

**DOI:** 10.3390/ijms252312651

**Published:** 2024-11-25

**Authors:** Anastasia V. Blokhina, Alexandra I. Ershova, Anna V. Kiseleva, Evgeniia A. Sotnikova, Anastasia A. Zharikova, Marija Zaicenoka, Yuri V. Vyatkin, Vasily E. Ramensky, Vladimir A. Kutsenko, Elizaveta V. Garbuzova, Mikhail G. Divashuk, Olga A. Litinskaya, Maria S. Pokrovskaya, Svetlana A. Shalnova, Alexey N. Meshkov, Oxana M. Drapkina

**Affiliations:** 1National Medical Research Center for Therapy and Preventive Medicine, Ministry of Healthcare of the Russian Federation, Petroverigsky per. 10, Bld. 3, 101000 Moscow, Russia; alersh@mail.ru (A.I.E.); sanyutabe@gmail.com (A.V.K.); sotnikova.evgeniya@gmail.com (E.A.S.); azharikova89@gmail.com (A.A.Z.); marija.zaicenoka@gmail.com (M.Z.); vyatkin@gmail.com (Y.V.V.); ramensky@gmail.com (V.E.R.); vlakutsenko@yandex.ru (V.A.K.); vostryakova.elizaveta@gmail.com (E.V.G.); divashuk@gmail.com (M.G.D.); olitinskaya@gnicpm.ru (O.A.L.); mpokrovskaia@list.ru (M.S.P.); sshalnova@gnicpm.ru (S.A.S.); meshkov@lipidclinic.ru (A.N.M.); drapkina@bk.ru (O.M.D.); 2Faculty of Bioengineering and Bioinformatics, Lomonosov Moscow State University, 1-73, Leninskie Gory, 119991 Moscow, Russia; 3Moscow Center for Advanced Studies, 20 Kulakova Str., 123592 Moscow, Russia; 4Department of Natural Sciences, Novosibirsk State University, 1 Pirogova Str., 630090 Novosibirsk, Russia; 5Institute for Artificial Intelligence, Lomonosov Moscow State University, 1-73, Leninskie Gory, 119991 Moscow, Russia; 6All-Russia Research Institute of Agricultural Biotechnology, 127550 Moscow, Russia; 7National Medical Research Center for Cardiology, 3–ya Cherepkovskaya Str., 15A, 121552 Moscow, Russia; 8Research Centre for Medical Genetics, 1 Moskvorechye Str., Moscow 115522, Russia; 9Department of General and Medical Genetics, Pirogov Russian National Research Medical University, 1 Ostrovityanova Str., 117997 Moscow, Russia

**Keywords:** familial dysbetalipoproteinemia, hyperlipoproteinemia type III, *APOE*, apolipoprotein E, dyslipidemia, autosomal dominant, pathogenicity, phenotype/genotype correlation, remnant lipoproteins, triglycerides

## Abstract

Familial dysbetalipoproteinemia (FD) is a highly atherogenic, prevalent genetically based lipid disorder. About 10% of FD patients have rare *APOE* variants associated with autosomal dominant FD. However, there are insufficient data on the relationship between rare *APOE* variants and FD. Genetic data from 4720 subjects were used to identify rare *APOE* variants and investigate their pathogenicity for autosomal dominant FD. We observed 24 variants in 86 unrelated probands. Most variants were unique (66.7%). Five identified *APOE* variants (p.Glu63ArgfsTer15, p.Gly145AlafsTer97, p.Lys164SerfsTer87, p.Arg154Cys, and p.Glu230Lys) are causal for autosomal dominant FD. One of them (p.Lys164SerfsTer87) was described for the first time. When we compared clinical data, it was found that carriers of pathogenic or likely pathogenic *APOE* variants had significantly higher triglyceride levels (median 5.01 mmol/L) than carriers of benign or likely benign variants (median 1.70 mmol/L, *p* = 0.034) and variants of uncertain significance (median 1.38 mmol/L, *p* = 0.036). For the first time, we estimated the expected prevalence of causal variants for autosomal dominant FD in the population sample: 0.27% (one in 619). Investigating the spectrum of *APOE* variants may advance our understanding of the genetic basis of FD and underscore the importance of *APOE* gene sequencing in patients with lipid metabolism disorders.

## 1. Introduction

The early development of atherosclerosis is often associated with inherited hyperlipidemias, which have various genetic bases [1]. Nowadays, genetic testing is becoming more available, and next-generation sequencing (NGS) makes it possible to investigate the genetic basis of inherited diseases more fully. This contributes to a broader study of inherited lipid metabolism disorders in various populations [2,3,4,5,6], including the Russian [7,8,9,10], and covers them at the global level [1,11,12].

Notably, when discussing monogenic hyperlipidemia, familial hypercholesterolemia (FH) is traditionally the focus of contemporary guidelines [1]. In contrast, familial dysbetalipoproteinemia (FD), which is also the most atherogenic and prevalent (from 0.2% to 2.7% [4,10,13,14]) hyperlipidemia, remains largely underestimated, underdiagnosed, and undertreated worldwide [10,15,16].

FD (OMIM #617347) has a complex multifactorial phenotype caused by the interaction of *APOE* gene variants with additional factors such as overweight or obesity [17,18,19], insulin resistance [17,18,20], diabetes mellitus [17,21], hypothyroidism, and so forth. As a result, elevated levels of cholesterol-enriched remnant lipoproteins contribute to the development of premature coronary [22,23,24] and peripheral [25,26] atherosclerosis. Therefore, early detection of predisposition to FD is essential to prevent cardiovascular events. Investigating the spectrum of *APOE* variants is key to advancing our understanding of the genetic basis of highly atherogenic FD.

*APOE* has three main alleles, ε2, ε3, and ε4, which form three homozygous (ε2ε2, ε3ε3, and ε4ε4) and heterozygous (ε2ε3, ε3ε4, and ε2ε4) genotypes. In over 90% of FD cases, the *APOE* ε2ε2 genotype predisposes to the development of the disease [15]. Therefore, most studies have focused specifically on the ε2ε2 genotype [4,14,19,21,27]. However, about 10% of FD patients have rare *APOE* variants [28]. These variants are inherited in an autosomal dominant trait. To date, only about 30 *APOE* variants associated with autosomal dominant FD have been reported [29,30]. The assessment of the pathogenicity of these variants for FD has either not been performed or has not been fully presented in publications [30].

Thus, in the present study, we aimed to investigate the spectrum of rare *APOE* variants in a large genetic sample and their pathogenicity for autosomal dominant FD. We also analyzed the prevalence of all identified *APOE* variants and estimated the expected prevalence of pathogenic or likely pathogenic *APOE* variants for autosomal dominant FD in the population sample.

## 2. Results

The graphical summary of the study process is presented in Figure 1.

### 2.1. Identification and Clinical Interpretation of Rare APOE Variants

Genetic data from 4720 subjects were analyzed to identify rare *APOE* variants. A total of 24 variants were detected as potentially causal for autosomal dominant FD in 86 unrelated probands (Figure 2, Table A1). Most of these variants were unique (66.7%). Eight variants were found in more than one proband. The vast majority of identified variants (83.3%) were located in exon 4. The most common variant type was missense (87.5%), while the rest were frameshifts.

Nine of the 24 identified variants were previously associated with FD. However, we considered only three of them as causal for autosomal dominant FD. Thus, frameshifts p.Glu63ArgfsTer15 and p.Gly145AlafsTer97 were predicted by Combined Annotation Dependent Depletion (CADD) as deleterious and classified as pathogenic. In addition, the missense variant p.Arg154Cys, located in the critical *APOE* receptor binding site, was detected in ten subjects. Among all identified *APOE* missense variants, p.Arg154Cys was predicted by CADD to be the most deleterious and classified as likely pathogenic for FD. The pathogenicity of four *APOE* variants (p.Gly145Asp, p.Gly183Ala, p.Glu262Lys, and p.Glu263Lys) previously described to cause autosomal dominant FD remains uncertain. p.Val254Glu and p.Arg269Gly were classified as benign. The total allele frequency of these variants was greater than 0.01% in gnomAD, and the FD lipoprotein phenotype was not confirmed in proband and relatives in a previous study [31].

We also detected five *APOE* variants previously associated with other hyperlipidemias but not with FD. One of them, p.Glu230Lys, was classified as likely pathogenic for FD in the current study. Three variants (p.Arg33His, p.Pro102Arg, and p.Ser314Arg) had an uncertain impact on the development of FD. p.Leu46Pro was the most prevalent *APOE* variant (38.4% of all probands), predicted to be tolerated, and classified as likely benign.

Four *APOE* variants (p.Arg50His, p.Gly138Ser, p.Ala184Val, and Thr307Ile) were registered in dbSNP but not reported in the literature. The pathogenicity of these variants for autosomal dominant FD remains uncertain.

Moreover, six novel *APOE* variants were described in this study. Among them, a frameshift variant in the *APOE* receptor binding site (p.Lys164SerfsTer87) was found to be pathogenic for FD. The variants p.Glu200Gly, p.Gly200Ala, p.Pro201Ser, and p.Val208Leu were of uncertain significance for FD.

### 2.2. APOE Genotypes in Carriers of Rare APOE Variants

Among carriers of rare *APOE* variants, ε3ε3 (56.9%) was the most prevalent genotype. This was followed by ε3ε4 (23.2%), ε2ε3 (7.0%), ε2ε4 (4.7%), and ε4ε4 (4.7%). The prevalence of the ε2ε2 genotype carriers was 3.5% (Table 1).

Rare *APOE* variants may combine with common *APOE* variants to form a unique rare *APOE* genotype. In our study, three carriers of the rare p.Gly145Asp variant had the ε2ε2 (homozygous for the common p.Arg176Cys variant), forming the rare ε2ε1 genotype [32,33,34,35,36]. In addition, three carriers of p.Leu46Pro had the ε4ε4 (homozygous for the common p.Cys130Arg variant), forming the Apoε4-Freiberg [37] (Table A2). Establishing the pathogenicity of rare *APOE* genotypes for FD or other dyslipidemias is complicated. Based on previous research data [32,33,34,35,36], we considered the ε2ε1 genotype as likely pathogenic for FD.

### 2.3. Prevalence of Rare APOE Variants

The prevalence of all identified rare *APOE* variants and the expected prevalence of pathogenic or likely pathogenic *APOE* variants associated with autosomal dominant FD were estimated in the ESSE-Ivanovo population-based sample (*n* = 1858) (Figure 3).

A total of 11 rare *APOE* variants were identified in 29 subjects from the ESSE-Ivanovo sample. Of these, three variants were pathogenic or likely pathogenic for FD (p.Gly145AlafsTer97, p.Lys164SerfsTer87, and p.Arg154Cys). Thus, the expected prevalence of causal variants for autosomal dominant FD in the population sample was 0.27% (one in 619) (95% CI: 0.09–0.63).

### 2.4. Carriage of Multiple Rare Variants

Among 86 probands with *APOE* variants, we observed eleven carriers (12.8%) of multiple rare variants (Table A3). Two of them were compound heterozygous for two *APOE* variants, one compound heterozygous for two *APOE* variants and homozygous for the *LMF1* variant of uncertain significance, seven double heterozygous for *APOE* variants and pathogenic *LDLR* variants, one double heterozygous with a missense variant in *APOE* and a pathogenic frameshift variant in *GPIHPB1*, and one compound heterozygous with two likely pathogenic variants in *LPL*. Thus, double heterozygosity involving *APOE* and *LDLR* pathogenic variants was the most prevalent (77.8% of all cases: *APOE*/*LDLR*, *APOE*/*LMF1*, *APOE*/*LPL*, and *APOE*/*GPIHPB1*).

### 2.5. Clinical Features Depending on APOE Variants’ Pathogenicity

We compared clinical parameters, including lipid levels, in carriers of *APOE* variants based on variant pathogenicity. For this purpose, the heterozygous carriers of *APOE* variants were divided into three groups: carriers of pathogenic or likely pathogenic variants, carriers of variants of uncertain significance, and carriers of benign or likely benign variants.

The group of pathogenic or likely pathogenic variants (*n* = 16) comprised carriers of pathogenic frameshifts p.Glu63ArgfsTer15, p.Gly145AlafsTer97, and p.Lys164SerfsTer87 (*n* = 3), likely pathogenic missenses p.Arg154Cys (*n* = 9) and p.Glu230Lys (*n* = 1), and carriers of a rare likely pathogenic ε2ε1 *APOE* genotype (*n* = 3).

All carriers of p.Arg50His, p.Gly138Ser, p.Gly183Ala, p.Gly200Glu, p.Val208Leu, p.Gln271Pro, p.Thr307Ile, and p.Ser314Arg, along with two carriers of p.Gly145Asp with ε3e3 genotype, were included as variants of uncertain significance (*n* = 13).

The group of benign or likely benign variants (*n* = 15) consisted of carriers of p.Val254Gly (*n* = 5), p.Arg269Gly (*n* = 2), and p.Leu46Pro with ε3ε3 genotype (*n* = 8).

In total, 44 carriers of rare *APOE* variants were included in the clinical data comparison (Table 2).

In the simultaneous analysis, three groups were comparable by age, sex, and atherosclerosis risk factors such as smoking, hypertension, and diabetes. The frequency of coronary heart disease was also comparable between groups. At the same time, the percentage of subjects taking lipid-lowering therapy (statins ± ezetimibe) was highest in carriers of pathogenic or likely pathogenic variants (56.3%), which was significantly different from carriers of variants of uncertain significance (7.7%, *p* = 0.024) but comparable to carriers of benign or likely benign variants (28.6%, *p* = 0.326).

Total cholesterol (TC), low-density lipoprotein cholesterol (LDL-C), high-density lipoprotein cholesterol (HDL-C), and non-high-density lipoprotein cholesterol (non-HDL-C) levels in carriers of causal variants for FD were comparable to those in carriers of variants of uncertain significance (*p* = 0.088, *p* = 0.598, *p* = 0.768, and *p* = 0.138, respectively) and carriers of benign or likely benign variants (*p* = 0.250, *p* = 0.598, *p* = 0.768, and *p* = 0.421, respectively). Notably, carriers of pathogenic or likely pathogenic *APOE* variants had significantly higher TG levels than carriers of benign or likely benign variants (*p* = 0.034) and variants of uncertain significance (*p* = 0.036) (Figure 4).

Table A4 summarizes the clinical characteristics of all heterozygous carriers of *APOE* variants, grouped by variants and genotypes.

## 3. Discussion

The involvement of the *APOE* gene in crucial human diseases is currently under active investigation [29,38,39]. Hyperlipidemia is no exception [5,6,10,29,35,40]. This is not surprising since *APOE* encodes apolipoprotein E, one of the most important links in lipid metabolism [41]. The interaction of *APOE* gene variants with additional factors contributes to the accumulation of cholesterol-enriched remnant lipoproteins. Finally, the highly atherogenic lipid disorder, namely FD, is developed [17,18,19,20,21,42].

The present study investigated the spectrum of rare *APOE* variants in a large genetic sample. Establishing the causal role of *APOE* variants to FD is complicated due to the multifactorial etiology of this disease [30]. To this point, we do not have specific related guidelines for *APOE* variant classification or FD-specific classification. Therefore, we analyzed the pathogenicity of *APOE* variants based on the American College of Medical Genetics and Genomics and the Association for Molecular Pathology (ACMG/AMP2015) guidelines [43], combined with the modern predictive metric [44] and comprehensive literature data analysis.

### 3.1. Causality Between Rare APOE Variants and Autosomal Dominant FD

Overall, we observed 24 rare *APOE* variants. The causality with autosomal dominant FD is not in doubt for only five of these variants. Frameshifts p.Glu63ArgfsTer15 and p.Gly145AlafsTer97 were previously described by our team as likely pathogenic for FD [10]. Notably, when we applied the CADD metric in the current study, these frameshifts were reclassified as pathogenic. p.Arg154Cys was also previously associated with FD. This variant is located in the critical *APOE* domain [45,46] and alters the amino acid residue, in which another variant, p.Arg154Ser, was found to be likely pathogenic for FD [5,29,47]. Most previous studies analyzed p.Arg154Cys only in carriers of ε2ε3 or ε2ε4 genotypes. For instance, Walden et al. [46] confirmed the FD phenotype by electrophoresis in heterozygous p.Arg154Ser and p.Arg176Cys carriers and showed in vitro that abnormal cell binding contributes to dysbetalipoproteinemia. Furthermore, Feussner et al. [45] demonstrated the segregation of p.Arg154Cys with FD in three generations (the proband and his father with ε2ε3 and two children with ε2ε3 and ε2ε4, respectively). Thus, in combination with p.Arg176Cys, p.Arg154Cys could be considered pathogenic for FD. In our study, p.Arg154Cys was detected only in carriers of the ε3ε3 genotype and was classified as likely pathogenic. p.Glu230Lys, previously observed in patients with familial combined hyperlipidemia [29], was classified as likely pathogenic for FD in the current report. In addition, one of the six novel *APOE* variants described in this study, p.Lys164SerfsTer87, was found to be pathogenic for FD.

Furthermore, one of the most notable identified missense variants is p.Gly145Asp. When p.Gly145Asp is not combined with the common p.Arg176Cys on the same allele, its causal role in FD is unclear. The opposite situation forms a unique ε1 isoform [32,33,34,35,36]. Thus, we detected three carriers of a rare ε2ε1 *APOE* genotype. It should be clarified that all these patients had a heterozygous p.Gly145Asp variant and a homozygous p.Arg176Cys variant. Therefore, evaluation of cis- or trans-position of these variants was not necessary in this case. It is difficult to know the deleterious effect of rare *APOE* variants on apolipoprotein E when they form a unique genotype. The effect could be additive, neutral, or could even offset the common variant’s dysfunction [48]. Hence, based on the literature data, we considered ε2ε1 as likely pathogenic for FD [32,33,34,35,36]. Integrating genotype information into future *APOE*-specific guidelines may improve *APOE* variant classification.

In addition, we classified common p.Leu46Pro as likely benign for FD. In particular, in the Zaheda 2014 study, there was no association between p.Leu46Pro and TG levels in subjects from US and African populations [49]. Nevertheless, p.Leu46Pro has been described in patients with FH and familial combined hyperlipidemia [6,40]. Besides, when p.Leu46Pro is combined with the common p.Cys130Arg, it forms a rare Apoε4-Freiberg genotype. This genotype has been previously studied with Alzheimer’s disease [38,50].

### 3.2. Carries of Multiple Rare Variants

Double heterozygosity involving *APOE* and *LDLR* pathogenic variants was the most common in this study. Carriage of multiple rare variants determines the complex interindividual variability between genotype and clinical phenotypes. In the Marmontel 2023 study, the lipid levels and the frequency of premature atherosclerotic cardiovascular diseases were compared between the *LDLR + APOE* carriers (*n* = 21) and the carriers of the same *LDLR* causative variants alone (*n* = 22) [40]. An additive effect of deleterious *APOE* variants on the FH phenotype was observed. Among patients with causal *APOE* variants, LDL-C levels were 46.0% higher in *LDLR* + *APOE* carriers than in *LDLR* carriers (mean 10.83 ± 3.45 versus 7.43 ± 1.59 mmol/L, *p* = 0.027), and premature atherosclerotic cardiovascular disease was more frequent (70.0% versus 30.0%, *p* = 0.026). Notably, there were no differences in TG levels (mean 1.46 ± 0.58 versus 1.36 ± 0.62 mmol/L, *p* = 0589) [40]. The results underscore the significance of *APOE* gene sequencing in patients with hyperlipidemia.

### 3.3. Prevalence of Pathogenic or Likely Pathogenic APOE Variants Associated with Autosomal Dominant FD

To date, numerous population-based studies have focused on the prevalence of autosomal recessive FD [4,10,13,14]. To our knowledge, this is the first study to estimate the prevalence of causal variants for autosomal dominant FD. The prevalence of these variants was considered as expected because our analysis included pathogenic or likely pathogenic variants identified in the current study. With the discovery of new pathogenic *APOE* variants and the review of the pathogenicity of variants of uncertain significance, the prevalence may increase in the future.

### 3.4. Clinical Features in Carriers of Causal Variants for Autosomal Dominant FD

Previous studies have shown that a TG cutoff of ≥1.5 mmol/L is optimal for identifying carriers of the ε2ε2 genotype and FD [10,51]. In our study, carriers of pathogenic or likely pathogenic *APOE* variants had a moderate increase in TG levels (median 5.01 mmol/L), which was significantly higher compared to other groups. Interestingly, a high PRS for TG levels was comparable in *APOE* variant carriers. This finding allowed us to exclude a polygenic effect on TG levels. In addition, we found a large variability in TG levels among causal variant carriers, ranging up to 19.31 mmol/L. At the same time, three causal variant carriers had TG levels less than 1.5 mmol/L. Moreover, pretreatment non-HDL-C levels were also high in causal variant carriers, underscoring their high cardiovascular risk.

All carriers of the ε2ε1 genotype had cutaneous eruptive and tendon xanthomas. A carrier of p.Glu63ArgfsTer15 had only tendon xanthomas, and a carrier of p.Arg154Cys had the eruptive xanthomas.

It should be emphasized that FD has a complex multifactorial phenotype and a delayed onset. Therefore, the diagnosis of FD is complicated. It should be based on a comprehensive analysis of genetic data and confirmation of the FD phenotype.

## 4. Materials and Methods

### 4.1. Sampling

Targeted (*n* = 4526) or exome (*n* = 194) sequencing data from three samples were analyzed to identify rare *APOE* variants.

The ESSE-Ivanovo population sample consisted of subjects from the Ivanovo region (median age was 48 years (37; 56), and 36.5% were men, *n* = 1858; Appendix A), selected from the “Epidemiology of Cardiovascular Diseases and Risk Factors in Regions of the Russian Federation” (ESSE-RF) study, a cross-sectional study conducted across 13 regions of Russia from 2012 to 2013 [52]. The Ivanovo region belongs to the European region of Russia and is representative of similar regions. The ethnic composition of the Ivanovo region is 95.57% Russian [53].

The FH sample consisted of patients with a clinical diagnosis of definite or probable heterozygous FH from the ESSE-FH-RF study (*n* = 158) [8] or examined at the National Medical Research Center (NMRC) for Therapy and Preventive Medicine (Moscow, Russia) (*n* = 399) [7]. In all cases (*n* = 557), FH was diagnosed using the Dutch Lipid Clinic Network criteria [54].

The Russian patient sample (RPS) included subjects with diverse chronic non-communicable diseases, whose blood samples were collected at the Biobank of the NMRC for Therapy and Preventive Medicine (*n* = 2305) [55].

### 4.2. Clinical and Biochemical Data

Retrospective clinical data from the ESSE-Ivanovo, FH sample, and RPS were used in the current study. The data included age, sex, body mass index, smoking status, presence of hypertension, diabetes, hypothyroidism, and presence of CHD (including myocardial infarction and coronary revascularization). The history of CHD was based on medical records and was diagnosed according to current European clinical guidelines. Carotid and femoral atherosclerosis and xanthomas were considered when these data were available. The type and volume of lipid-lowering therapy was also analyzed.

Retrospective lipid levels, including TC, LDL-C, HDL-C, and TG, were reported in mmol/L. Non-HDL-C was calculated as TC minus HDL-C and reported in mmol/L. All lipid levels were previously measured using the Abbott Architect C-8000 system (Abbott Laboratories, North Chicago, IL, USA). In the current study, pretreatment lipid levels were presented. In the ESSE-Ivanovo and ESSE-FH-RF studies, LDL-C levels were determined directly. In most cases, the Friedewald formula was used to calculate LDL-C levels in RPS. Therefore, LDL-C levels from RPS were only presented for patients with TG levels < 4.5 mmol/L or those whose LDL-C was directly measured. For patients on regular statin therapy (*n* = 1), pretreatment LDL-C levels were estimated using the average relative decrease in concentration with the corresponding dose of atorvastatin [56]. Pretreatment TG levels were estimated using a correction factor for statin (*n* = 2) or fenofibrate (*n* = 1) from a local study [57]. TC and HDL-C levels were not recalculated in this case.

Carriers of rare *APOE* variants, without pathogenic variants in other genes associated with lipid metabolism disorders, not compound heterozygous for *APOE* variants, and with available clinical data were included for clinical data comparison. When *APOE* variants formed a rare *APOE* genotype, only carriers of ε2ε1 were included in the clinical data analysis.

### 4.3. Genetic Analysis

#### 4.3.1. DNA Extraction

Blood samples were stored at −32 °C at the Biobank of the NMRC for Therapy and Preventive Medicine [54]. Genomic DNA from peripheral blood was extracted using the QIAamp DNA Blood Mini Kit (Qiagen, Hilden, Germany). DNA concentration was measured using the Qubit 4.0 fluorimeter (Thermo Fisher Scientific, Waltham, MA, USA).

#### 4.3.2. Sequencing

NGS was performed with two platforms: NextSeq 550 (*n* = 4285: custom panel (*n* = 4091), exome (*n* = 194)) and Ion S5 (*n* = 435: custom panel). All sequencing steps were performed according to the manufacturers’ protocols.

For the NextSeq 550 platform (Illumina, San Diego, CA, USA), the paired-end sequencing (150 or 300 bp) was performed. For targeted sequencing (custom panel), libraries were prepared using the SeqCap EZ Prime Choice Library Kit (Roche, Basel, Switzerland). Exome libraries were prepared with the IDT-Illumina TruSeq DNA Exome protocol (Illumina, San Diego, CA, USA).

For the Ion S5 platform (Thermo Fisher Scientific, Waltham, MA, USA), the 200 bp sequencing was performed. The AmpliSeq libraries using custom panel were prepared on the Ion Chef System (Thermo Fisher Scientific, Waltham, MA, USA).

The set of 24 genes associated with dyslipidemia (*ABCA1*, *ABCG5*, *ABCG8*, *ANGPTL3*, *APOA1*, *APOA5*, *APOB*, *APOC2*, *APOC3*, *APOE*, *CETP*, *GPD1*, *GPIHBP1*, *LCAT*, *LDLR*, *LDLRAP1*, *LIPC*, *LIPI*, *LMF1*, *LPL*, *PCSK9*, *SAR1B*, *STAP1*, *USF1*) was analyzed [10].

Sanger sequencing was performed using the DNA sequencer Applied Biosystem 3500 Genetic Analyzer (Thermo Fisher Scientific, Waltham, MA, USA) and the ABI PRISM BigDye Terminator v3.1 reagent kit (Thermo Fisher Scientific, Waltham, MA, USA), following the manufacturer’s protocol.

#### 4.3.3. Bioinformatic Analysis and Clinical Interpretation

The GRCh37/hg19 reference genome was selected for aligning paired-end reads. A custom pipeline [58] based on GATK 3.8 [59] was used to process the sequencing data and evaluate quality control. VCF files were then generated, containing a list of variants, their genomic coordinates, coverage data, and other characteristics. Low-quality variants, likely due to sequencing errors, have been filtered out. The coverage depth of the reference and alternative alleles, the quality of reads and mapping, and other relevant factors were reported and analyzed.

Variants were described according to the Human Genome Variation Society recommendations (HGVS; https://hgvs-nomenclature.org/stable/, accessed on 1 August 2024). cDNA was numbered from +1 for A in the ATG translation initiation codon of the reference sequence (NM_000041.4). Amino acid residues were numbered from +1 for the initiating methionine of the protein sequence (NP_000032.1). The signal peptide (i.e., the first 18 amino acids of apolipoprotein E) was included, and the transcript of *APOE* (NM_000041.4) consisted of 317 amino acids. The difference is therefore +18 to the original nomenclature and -26 to the NM_001302688.1 transcript.

Non-synonymous variants with a minor allele frequency (MAF) <0.5% across populations in the Genome Aggregation Database (gnomAD v2.1.1; http://gnomad.broadinstitute.org, accessed on 1 August 2024) or those missing gnomAD were selected. Variant annotation was performed using OMIM [60], gnomAD (v2.1.1) [61], ClinVar [62], Human Gene Mutation Database (HGMD) [63], Leiden Open Variation Database (LOVD) [64], dbSNP [65] databases, and literature data, including segregation information.

Clinical interpretation of *APOE* variants was based on the ACMG/AMP2015 guidelines [43]. Combined Annotation Dependent Depletion (CADD v1.7 for GRCh37) was used to predict the impact of single nucleotide *APOE* variants as well as deletion variants [44]. CADD integrates multiple annotations into one metric by contrasting variants that survived natural selection with simulated mutations [44]. Scaled CADD scores (PHRED-like scaled C-scores) were obtained to access the deleteriousness of *APOE* variants [66]. Variants with a PHRED score greater than or equal to 20 (predicted to be the 1% most deleterious substitutions of all GRCh37/hg19 reference single nucleotide variants) were considered as potentially pathogenic. In this case, criterion PP3 ACMG/AMP2015 was used [66]. Variants with a PHRED score of less than 10 were considered potentially benign. Therefore, criterion BP4 ACMG/AMP2015 was applied [66].

For *APOE* variant interpretation, the autosomal dominant FD phenotype was based on the available literature data. We used cut-offs from lipoprotein ultracentrifugation, electrophoresis, or well-established biochemical algorithms [30]. When at least one of these criteria was met, criterion PP4 ACMG/AMP2015 was applied. Otherwise, we applied criterion BP6.

Common *APOE* genotypes (ε3ε3, ε4ε4, ε2ε2, ε2ε3, ε3ε4, and ε2ε4) were identified as previously described [10]. Information on cis- or trans-position of *APOE* variants was not available for evaluation. Therefore, we presented rare *APOE* genotypes only for patients with common homozygous *APOE* genotypes. Only established and known cases of rare *APOE* genotypes from the published literature were presented in the current study.

### 4.4. Polygenic Risk Score

The weighted polygenic risk score (PRS) was calculated using the β-coefficients from the original article, which previously demonstrated significant associations with TG levels (40 variants predicting TG levels [67]) in the population of the European part of Russia. PRS of the study participants was compared with that of all 1858 subjects from the population-based sample (ESSE-Ivanovo). A high PRS of hypertriglyceridemia was defined as a weighted PRS >80th percentile, whereas a low PRS was indicated by a weighted PRS <50th percentile. A high polygenic contribution was represented by a range of weighted PRS from (−1.213) to (−1.841) in the ESSE-Ivanovo sample. Values from (−2.037) to (−2.820) were set for a low contribution (Appendix A). PRS for TG was calculated for all study participants, except two patients with FH.

### 4.5. Prevalence of Rare APOE Variants Associated with Autosomal Dominant FD

The prevalence of identified rare *APOE* variants was investigated in the ESSE-Ivanovo population-based sample (*n* = 1858). To estimate the expected prevalence of pathogenic or likely pathogenic *APOE* variants associated with autosomal dominant FD, we summarized the frequencies of these variants identified in the ESSE-Ivanovo sample.

### 4.6. Ethical Statement

The study was conducted in accordance with the Declaration of Helsinki and the National Standard of the Russian Federation “Good Clinical Practice (GCP)” GOST R52379-2005 and was approved by the Independent Ethics Committee of the National Medical Research Center for Therapy and Preventive Medicine (protocol number 07-05/20 dated 26 November 2020). In order to comply with the above-mentioned laws, as well as Article 93 of the Federal Law “On the Fundamentals of Health Protection of Citizens of the Russian Federation” dated 21 November 2011, No. 323-FZ, each subject signed a written consent to the processing of their personal data. Data from the ESSE-Ivanovo study, the ESSE-FH-RF study, and the RPS cohort were used in the current study. Written informed consent was obtained from each patient as part of their participation in these scientific projects. ESSE-Ivanovo, ESSE-FH-RF, and RPS data were accessed from 27 November 2020. The database containing clinical, biochemical, and genetic data was de-identified and encrypted to ensure confidentiality.

### 4.7. Statistical Analysis

Statistical analyses were performed using R version 4.3.2 (R Foundation for Statistical Computing, Vienna, Austria) [68]. Continuous variables were summarized as median (Me) and interquartile range (Q1; Q3), while categorical variables were presented as absolute numbers and percentages. To compare continuous variables between two independent groups, the Mann-Whitney U test was used, and for categorical variables, the two-sided Fisher’s exact test was used. For comparisons between three or more independent groups, the Kruskal–Wallis test was used for continuous variables and the two-sided Fisher’s exact test for categorical variables. TG levels were logarithmized for comparison of clinical data. In this case, the t-test was used to compare TG levels of two independent groups. The p-values from pairwise comparisons were adjusted using the Holm–Bonferroni method.

We calculated the prevalence of all rare *APOE* variants and the expected prevalence of pathogenic or likely pathogenic *APOE* variants associated with autosomal dominant FD by dividing the number of subjects with these parameters by the total sample size. The prevalence was calculated as a percentage for all participants. The Clopper–Pearson exact method was used for the estimation of the 95% confidence interval.

A *p*-value of less than 0.05 was considered statistically significant. Data visualization was carried out using the ggplot2 package [69].

### 4.8. Limitations

Our study has several limitations. First, we did not use lipoprotein ultracentrifugation or electrophoresis to confirm the FD phenotype as part of the assessment of the pathogenicity of *APOE* variants. Also, we could not apply previously developed FD diagnostic algorithms, including apoB level, in Russian patients, as we previously described [10]. Instead, the autosomal dominant FD phenotype was based on the cut-offs of lipoprotein ultracentrifugation, electrophoresis, or well-established biochemical algorithms from literature data when available for the *APOE* variant analyzed. Second, information on cis- or trans-position of *APOE* variants was not available for evaluation. Therefore, we presented rare *APOE* genotypes only for carriers of common homozygous *APOE* genotypes. For the same reason, patients with the unclear genotype were excluded from the clinical data comparison.

## 5. Conclusions

This study provides a comprehensive clinical interpretation of rare *APOE* variants associated with highly atherogenic autosomal dominant FD. We identified 24 *APOE* variants, five of which were causal for autosomal dominant FD, including the novel frameshift variant p.Lys164SerfsTer87. Our findings also showed that carriers of pathogenic or likely pathogenic *APOE* variants have higher TG levels than carriers of benign or likely benign variants and variants of uncertain significance. Furthermore, we have estimated for the first time the expected prevalence of causal variants for autosomal dominant FD in a population-based sample.

Investigating the spectrum of *APOE* variants may advance our understanding of the genetic basis of FD and underscore the important role of *APOE* gene sequencing in patients with lipid metabolism disorders. We believe that determining the contribution of these variants to the development of autosomal dominant FD may facilitate earlier diagnosis, timely treatment, and improved prevention of highly atherogenic FD.

## Figures and Tables

**Figure 1 ijms-25-12651-f001:**
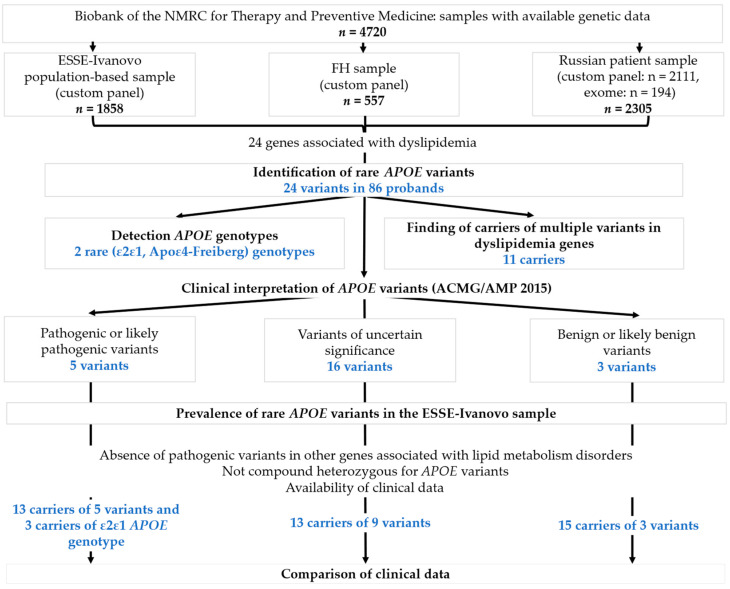
The study design. FD—familial dysbetalipoproteinemia; FH—familial hypercholesterolemia.

**Figure 2 ijms-25-12651-f002:**
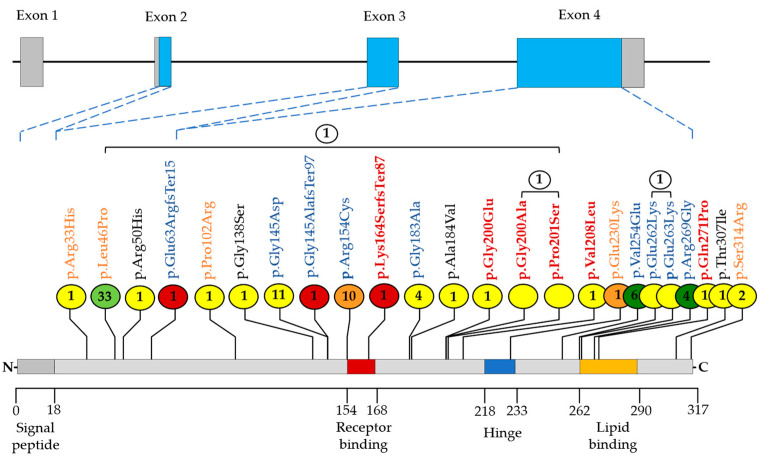
Spectrum of rare *APOE* variants identified in the study. The N-terminal signal peptide and three major *APOE* regions are shown: the N-terminal region containing the receptor binding site; the C-terminal region, which contains the lipid binding site; and a hinge region connecting the N- and C-terminal regions. The number of index patients (*n* = 86) is indicated in the circle. The number of carriers of two rare *APOE* variants is shown in white circles. Black brackets indicate the *APOE* variants in these compound heterozygotes. The color of the circle indicates the clinical interpretation: red, orange, yellow, light green, and green for pathogenic, likely pathogenic, variant of uncertain significance, likely benign, and benign variants, respectively. Novel variants are highlighted in red, previously associated with autosomal dominant FD; in blue, with other hyperlipidemias; in orange, registered in dbSNP but not published; in black.

**Figure 3 ijms-25-12651-f003:**
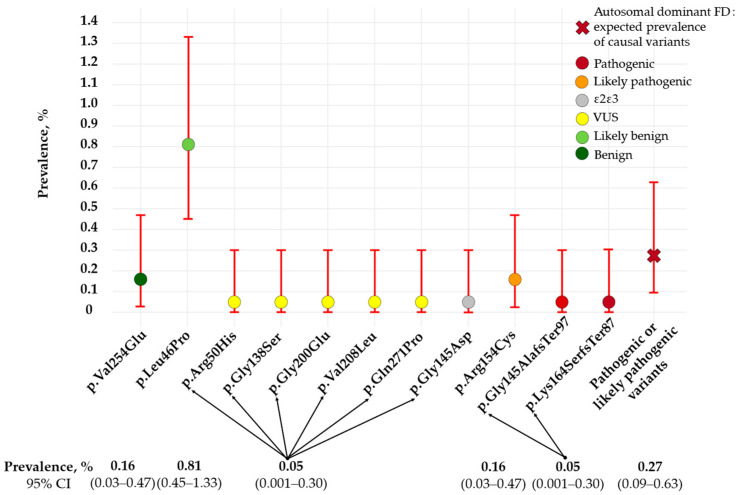
Prevalence of rare *APOE* variants and the expected prevalence of pathogenic or likely pathogenic *APOE* variants associated with autosomal dominant FD in the ESSE-Ivanovo sample. The color of the circle indicates the pathogenicity of the variant.: red, orange, yellow, light green, and green for pathogenic, likely pathogenic, variant of uncertain significance, likely benign, and benign variants, respectively. Gray indicates carrier of p.Gly145Asp variant with ε2ε3 genotype (information on cis- or trans-position of this variant was not available for evaluation). VUS—variant of uncertain significance.

**Figure 4 ijms-25-12651-f004:**
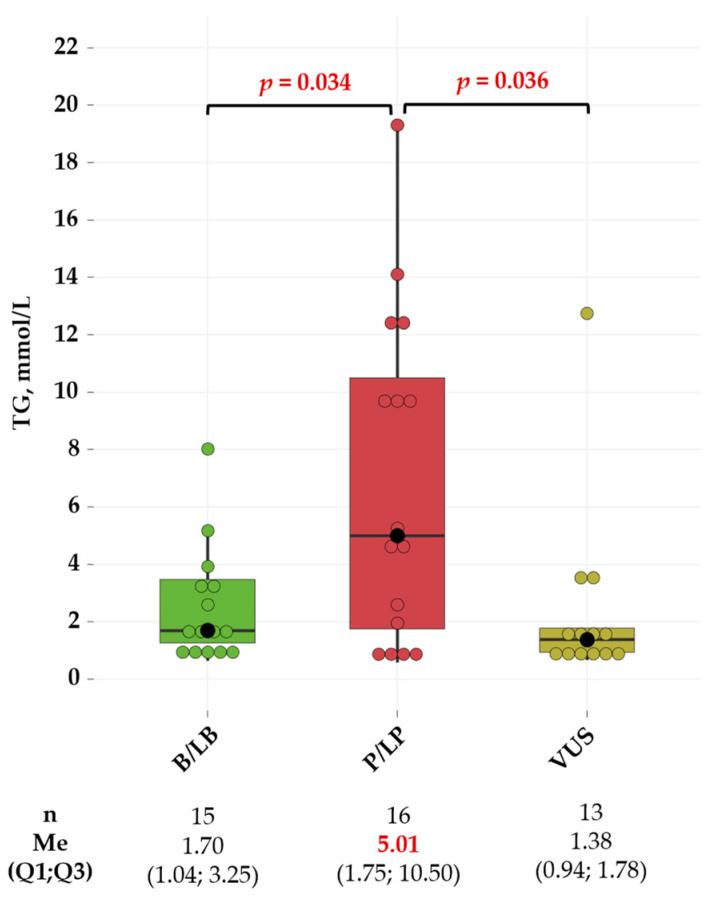
TG levels of carriers of pathogenic or likely pathogenic variants compared to carriers of benign or likely benign variants and carriers of variants of uncertain significance. Central lines represent the median, box limits represent upper and lower quartiles, vertical lines represent 1.5 times the quartile range, and individual data points outside this range are shown as outliers. The colored points represent carriers of *APOE* variants, highlighting their specific TG levels within the overall distribution (green for carriers of benign or likely benign variants, red for carriers of pathogenic or likely pathogenic variants, and yellow for carriers of variants of uncertain significance). *t*-test was used for pairwise comparison of TG levels. *p*-values were adjusted by the Holm–Bonferroni method. Only significant differences are indicated among groups. B—benign; LB—likely benign; LP—likely pathogenic; Me—median; P—pathogenic; VUS—variant of uncertain significance.

**Table 1 ijms-25-12651-t001:** Prevalence of main *APOE* genotypes in patients with rare *APOE* variants.

Sample, *n*	*APOE* Genotype, *n* (%) (95% Confidence Interval)					
	ε3ε3	ε2ε2	ε4ε4	ε2ε3	ε3ε4	ε2ε4
86	49 (56.9) (45.85–67.61)	3 (3.5) (0.73–9.86)	4 (4.7) (1.28–11.48)	6 (7.0) (2.60–14.57)	20 (23.2) (14.82–33.61)	4 (4.7) (1.28–11.48)

**Table 2 ijms-25-12651-t002:** Clinical characteristics of heterozygous carriers of *APOE* variants, grouped by pathogenicity.

Parameter	All Carriers (*n* = 44)	Carriers of Pathogenic or Likely Pathogenic Variants (*n* = 16)	Carriers of VUS (*n* = 13)	Carriers of Benign or Likely Benign Variants (*n* = 15)	*p*-Value *
Men, *n* (%)	19 (43.2)	8 (50.0)	6 (46.2)	5 (33.3)	0.693
Age, years, Me (Q1; Q3)	51 (42; 57)	51 (40; 56)	48 (44; 55)	52 (43; 61)	0.878
Current smoking, *n* (%)	7 (16.3) *n* = 43	4 (25.0)	0	3 (21.4) *n* = 14	0.161
Ex-smokers, *n* (%)	9 (20.9) *n* = 43	2 (12.5)	5 (38.5)	2 (14.3) *n* = 14	0.207
Hypertension, *n* (%)	24 (54.5)	8 (50.0)	7 (53.8)	9 (60.0)	0.928
BMI, kg/m^2^, Me (Q1; Q3)	28.4 (24.5; 30.7) *n* = 43	28.5 (26.0; 31.5)	28.8 (25.5; 30.9) *n* = 12	25.0 (22.3; 29.0)	0.196
Diabetes, *n* (%)	8 (18.2)	4 (25.0)	2 (15.4)	2 (13.3)	0.705
Hypothyroidism, *n* (%)	1 (2.3)	0	0	1 (6.7)	0.636
CHD, *n* (%)	4 (9.1)	2 (12.5)	1 (7.7)	1 (6.7)	1.0
LLT, *n* (%)	14 (31.8) *n* = 43	9 (56.3) **	1 (7.7) **	4 (28.6) *n* = 14	**0.017**
TC, mmol/L, Me (Q1; Q3)	5.82 (4.53; 7.35) *n* = 40	7.10 (4.90; 15.89) *n* = 13	4.87 (3.98; 5.95) *n* = 12	6.37 (4.74; 7.20)	0.078
LDL-C, mmol/L, Me (Q1; Q3)	3.02 (2.46; 3.97)	2.94 (2.46; 3.81) n= 11	2.68 (2.07; 3.41)	3.86 (2.96; 4.26)	0.185
HDL-C, mmol/L, Me (Q1; Q3)	1.17 (0.96; 1.50) *n* = 43	1.14 (0.88; 1.40) n=15	1.17 (1.06; 1.51)	1.18 (1.03; 1.62)	0.644
Non-HDL-C, mmol/L, Me (Q1; Q3)	4.44 (3.19; 6.10) *n* = 39	5.29 (3.74; 12.47) *n* = 12	3.75 (2.50; 4.79) *n* = 12	5.10 (3.67; 5.83)	0.142
TG, mmol/L, Me (Q1; Q3)	1.84 (1.01; 4.86)	5.01 *** (1.75; 10.50)	1.38 *** (0.94; 1.78)	1.70 *** (1.04; 3.25)	0.061
ε3ε3, *n* (%)	38 (86.4)	12 (75.0)	11 (84.6)	15 (100)	0.114
ε2ε2, *n* (%)	3 (6.8)	3 (18.7)	0	0	0.098
ε2ε3, *n* (%)	1 (2.3)	1 (6.3)	0	0	1.0
ε3ε4, *n* (%)	2 (4.5)	0	2 (15.4)	0	0.082
High PRS TG, *n* (%)	10 (25.0) *n* = 40	4 (26.7) *n* = 15	3 (25.0) *n* = 12	3 (23.1) *n* = 13	1.0

Data are presented for patients without pathogenic variants in other genes associated with lipid metabolism disorders, not compound heterozygous for *APOE* variants, and with available clinical data. When *APOE* variants formed a rare *APOE* genotype, only patients with ε2ε1 were included in the clinical data analysis, and the genotype was considered likely pathogenic. * *p*-values indicate differences between three groups. The Kruskal–Wallis test was used for continuous variables and the two-tailed Fisher’s exact test for categorical variables. ** *p* < 0.05 for differences between carriers of pathogenic or likely pathogenic variants and carriers of variants of uncertain significance, obtained by pairwise comparisons using the two-tailed Fisher’s exact test. *p*-values were adjusted by the Holm–Bonferroni method. *** *p* < 0.05 for differences between carriers of pathogenic or likely pathogenic variants compared to carriers of benign or likely benign variants and carriers of variants of uncertain significance. *p*-values were obtained by pairwise comparisons using *t*-test and were adjusted by the Holm–Bonferroni method. BMI—body mass index; CHD—coronary heart disease; HDL-C—high-density lipoprotein cholesterol; LDL-C—low-density lipoprotein cholesterol; LLT—lipid-lowering therapy; Me—median; Non-HDL-C—non-high-density lipoprotein cholesterol; PRS—polygenic risk score; TC—total cholesterol; TG—triglycerides; VUS—variant of uncertain significance.

## Data Availability

The data used and/or analyzed during the current study are available from the corresponding authors on reasonable request. Individual genotype information cannot be made available in order to protect participant privacy.

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
