# Peer review of "Spectrum and Prevalence of Rare APOE Variants and Their Association with Familial Dysbetalipoproteinemia"

_ijms, 2024, doi:10.3390/ijms252312651_

Round 1
Reviewer 1 Report
Comments and Suggestions for Authors
The ms of Blokhina et al. is potentially interesting and can contribute to the understanding of rare APOE variants and their association to Familial Dysbetalipoproteinemia. Some points should be improved before publication:
1) There are many repetitions in the text, i.e. the sentence “two of them were frameshift ….and were previously described by our team [10]” is repeated 3 times. Please check all repetitions.
2) Results, paragraph “Clinical Data Analysis” is confused: at first there is a mention to patients, then to variants, then again to patients. The style is made of scattered sentences not linked one to another. Please rewrite making the logical connections among sentences clearer.
3) Figure 3 is based on Table 2: the authors should explain why the p values in the figure are different from that reported in the table and state to what each p value refers to.
4) Table 2 again and p. 7 lines 2-4 from top. The authors should also mention the difference in percentage among P/LP versus B/LB and not only versus VUS.
5) The Discussion is organized by a list with the subtitles of the paragraphs of the Results and globally is too long and not well merged. It should be rewritten and shortened.
Reviewer 2 Report
Comments and Suggestions for Authors
This topic is interesting and well described. anyhow it should be improved adding to Tab 2 the value of non-HDL as the LDL-C, if calculated could lead to inappropriate results.
Furthermore it is notable to observe the highest LDL-C values in the normal/likely benign ApoE variants. Should you discuss this point. For MD practice the lipid phenotype represents the first step to consider for further analysis, besides clinical signs
Reviewer 3 Report
Comments and Suggestions for Authors
Although the study is very interesting, it is unfortunately one of the most chaotic papers I have read recently. It is difficult to follow the results, it is not entirely clear which samples were analysed and how, and some of the tables are not properly presented and therefore difficult to read. With a better structure and clearer presentation it would be a great paper, but in this form I can't accept it at the moment and unfortunately I don't have the time to describe each point in detail.
Round 2
Reviewer 1 Report
Comments and Suggestions for Authors
The authors have answered to all points modifying the manuscript as requested.
Reviewer 3 Report
Comments and Suggestions for Authors
Thank you for the extensive revision of your manuscript. It is much improved and reads well now. Please be aware of a few typos when you receive it for proofreading.